# Research on VFTO Identification of GIS Based on Wavelet Transform and Singular Value Decomposition

Gang Xiao [1,2], Quansen Rong [1,*], Miaoran Yang [1], Peng Xiao [1,*], Qihong Chen [1], Junzhe Fan [1], Haoran Guo [1] and Haonan Wang [1]

[1] School of Automation, Wuhan University of Technology, Wuhan 430070, China; 290401@whut.edu.cn (G.X.); 302907@whut.edu.cn (M.Y.); chenqh@whut.edu.cn (Q.C.); 300374@whut.edu.cn (J.F.); 302984@whut.edu.cn (H.G.); 303075@whut.edu.cn (H.W.)
[2] Wuhan Digital Engineering Institute, Wuhan 430074, China
* Correspondence: 302825@whut.edu.cn (Q.R.); xp@whut.edu.cn (P.X.)

**Abstract:** The accurate identification of Very Fast Transient Overvoltage (VFTO) is the key of overvoltage control in modern smart grids. In order to accurately identify VFTO generated by the operation of a disconnector in Gas Insulated Substation (GIS), a VFTO identification method based on Wavelet Transform (WT) and Singular Value Decomposition (SVD) is proposed. The simulation model of VFTO is established in ATP-EMTP software first, and then wavelet decomposition is used in MATLAB software for VFTO simulation of the waveform from the ATP-EMTP software. Then, the feature matrix is composed of the coefficients of each frequency layer of the wavelet. The SVD is used to decompose the feature matrix, and finally the characteristic parameters of the VFTO are obtained. The simulation results in Matlab software indicate that the characteristic parameters of VFTO have an obvious difference compared with those of power frequency AC voltage, especially in the load-side, which verifies the effectiveness of the VFTO identification method based on WT and SVD proposed in this paper.

**Keywords:** VFTO; wavelet analysis; singular value; identification; characteristic parameters





## 1. Introduction

In recent years, the Gas Insulated Substation (GIS) is becoming more and more common in ultra-high voltage grids, where the disconnector can generate VFTO when opening and closing. Not only is VFTO harmful to the insulation of primary equipment, but the consequent electromagnetic interference can cause the misoperation of secondary equipment [1–4]. On the background of a smart grid, accurate identification of various overvoltage signals plays an important role in power grid risk assessment and fault detection. In order to reduce the harm of VFTO to power systems and electrical equipments, it is necessary to conduct in-depth research on its generation mechanism and waveform characteristics. Due to the high amplitude and steep wave front of VFTO, it is difficult to accurately measure its original waveform, and the accuracy will be decreased by the interference from environmental noise and other overvoltage signals while recognizing VFTO. Accurate identification of VFTO is of great significance for real-time monitoring of power system overvoltage, ensuring power supply reliability and improving power quality.

There is growing interest in the modeling, analysis, and suppression of VFTO [5–9]. The simulation analysis for different types of switch operation in 110 kV GIS substations was carried out by ATP-EMTP software, and the most influential type was found [5]. An analysis of VFTO simulations under the situation of a circuit breaker and disconnecting switch failure was given in [6]. An all-current Partial Element Equivalent Circuit (PEEC) model was established to analyze the impact of VFTO on Stepped Controlled Shunt Reactors (SCSR) [7]. The interaction between the Transient Enclosure Voltage (TEV) caused by VFTO and the grounding system was analyzed through simulation and actual verification [8].

However, VFTO can cause great harm to the insulation of electrical equipments, so it is necessary to reduce its amplitude and wave front steepness. The proposed methods include paralleling switch resistors, installing arresters, and optimizing the operation sequence of isolation switches. Moreover, a type of VFTO suppression method called the spiral tube damping busbar was given, and it has a conspicuous inhibitory effect on VFTO after further improvement [9].

In addition, the capacitance divider method and differential integration method are two common methods used to measure VFTO, which can accurately measure the frequency characteristics of VFTO. The research interest in the method of VFTO measurement is growing [10–13]. The proposed miniaturized measurement system in [10] can be applied in 500 kV GIS substations to measure VFTO. It could be well seen from the literature [11] that the main advantage of the proposed equivalent circuit of the differentiating-integrating VFTO measurement is in there being no need to modify GIS. Furthermore, a novel conical voltage sensor for VFTO measurement in ultra-high-voltage GIS was developed, which can reduce the refraction and reflection of VFTO [12]. A black-box approach using S-parameters was designed, expanding the VFTO measurement frequency ranging up to 50 MHz at GIS facilities [13].

The most-used methods for identifying fault voltage waveforms include Fourier transform, Gabor transform, Wigner–Will distribution, and wavelet transform. Fourier transform consists in transforming the original signal into the sum of several sinusoidal signals, and Gabor transform adds a fixed-size time window on the basis of Fourier transform; neither of them can effectively extract the characteristics of VFTO transient waveforms. On the contrary, wavelet transform has a variable-size time window, which can amplify the details of the transient waveform and is suitable for analyzing transient processes such as VFTO. Thus, some scholars focused on the identification of VFTO [14–19]. Several methods for the spectrum analysis of VFTO were compared, including continuous wavelet transform, short-time Fourier transform, Wigner–Will distribution, and smooth pseudo Wigner–Will distribution [14]. A type of arc fluid model was built to simulate the disconnector's operation and obtain the characteristics of VFTO waveforms [15]. Furthermore, a novel power transformer condition monitoring system (NCMS) that can capture VFTO was established, which also worked for core multi-point earth faults and core deep saturation faults [16]. A prediction model of V–t curves under VFTO and lightning impulse voltage was established, which can reflect the breakdown characteristics of long $SF_6$ gas gaps accurately [17]. In addition, the identification eigenvalues of lightning overvoltage, intermittent arc grounding overvoltage, and high-frequency resonant overvoltage were found using wavelet decomposition and SVD [18]. On the other hand, a high recognition accuracy method based on WT and SVD was proposed in [19] under different load conditions.

The wavelet analysis is adopted to better extract the characteristics of VFTO waveforms in this paper. Singular value decomposition is used to reduce the dimensions of data, which improves the efficiency of calculation. The rest of the paper is organized as follows. Section 2 centers on describing the structure of GIS. Section 3 puts forward a method of VFTO recognition based on WT and SVD. In Section 4, wavelet decomposition of VFTO voltage waveform is used, and the wavelet coefficients of each frequency layer are obtained. Next, the feature matrix is composed and the SVD is used to decompose the feature matrix, constructing singular value characteristic parameters and energy characteristic parameters for identifying VFTO. Finally, conclusions are drawn in Section 5.

## 2. Structure of GIS

The structure of a GIS substation is shown in Figure 1. The power frequency power supply is connected to the casing tube through the transformer. One side of the disconnector is connected to the casing tube, and the other side is connected to the load. The metal shell is filled with $SF_6$ gas insulating medium.

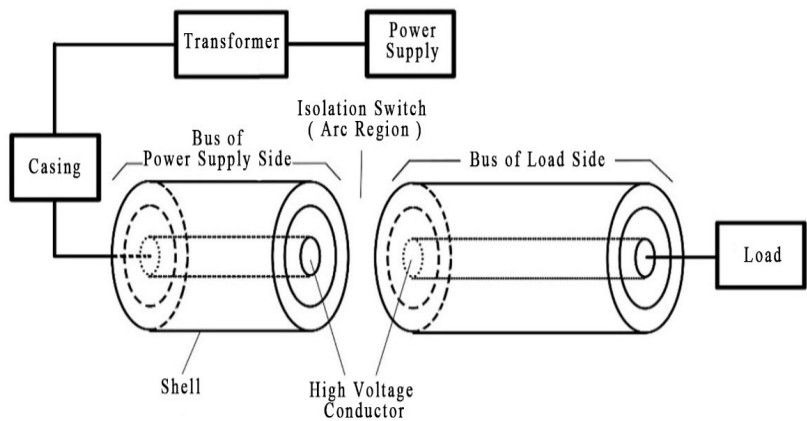

**Figure 1.** GIS line structure model diagram.

In the process of isolating the switch operation, the voltage difference between the dynamic and static contacts is greater than the insulation withstanding voltage of the $SF_6$ medium, the gap is broken down, and the arc is generated. When the rising speed of the recovery voltage between the two contacts after the arc extinguishing is greater than the insulation recovery strength of the $SF_6$ medium, arc reignition will occur, as shown in Figure 2.

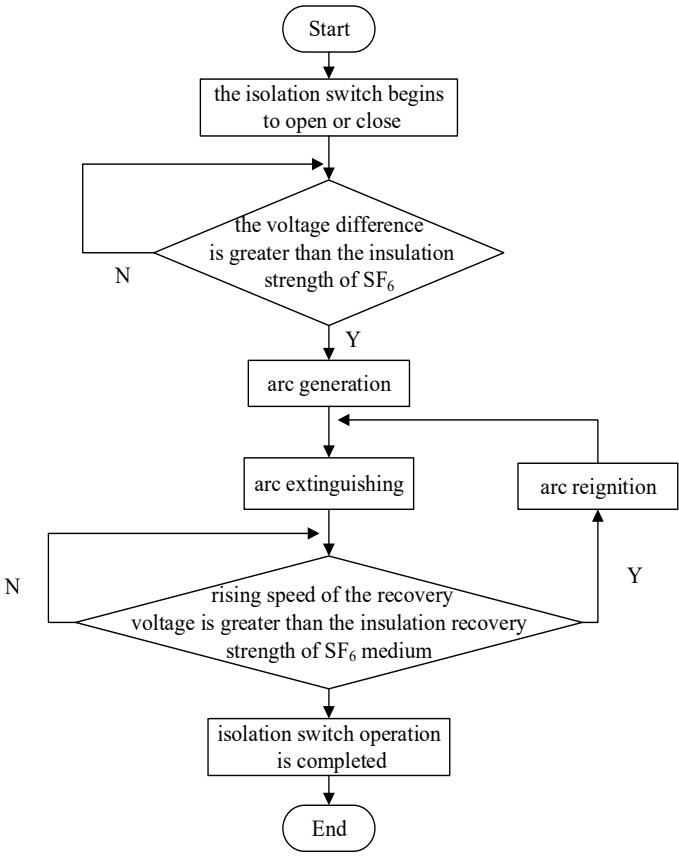

**Figure 2.** Arcing and reignition process in isolation switch operation.

The voltage wave generated in this process continuously reflects, refracts in the GIS substation line, and is superimposed to form VFTO. The basic oscillation frequency of VFTO is determined by the parameters of capacitance and inductance of the line, and the

high-frequency oscillation frequency is determined by the superposition of reflected and refracted voltage waves, which is related to the length of the bus.

### 3. VFTO Identification Method Based on WT and SVD

*3.1. Continuous Wavelet Transform of VFTO*

VFTO is represented by the function $f(t)$, which is transformed by wavelets to obtain its wavelet coefficient $\alpha(s, \delta)$, as shown in Equation (1).

$$\alpha(s, \delta) = \frac{1}{\sqrt{s}} \int f(t) \cdot \phi\left(\frac{t - \delta}{s}\right) dt \tag{1}$$

In Equation (1), $s$ is the contraction–expansion factor, which controls the extension scale of the wavelet. If $s$ increases, the waveform of the wavelet $\phi_{s,\delta}(t)$ stretches, and otherwise the waveform shrinks. $\delta$ is a translation factor. Changing the value of $\delta$ can make the wavelet waveform shift along the time axis. $\phi\left(\frac{t-\delta}{s}\right)$ is a wavelet cluster, which originates from the wavelet-generating function $\phi(t)$ through expansion and shift transformation.

Due to the high frequency of VFTO, a narrow time window is selected to amplify the local details of the original VFTO waveform, so the value of $s$ is very small. The Daubechies wavelet is a compactly supported orthogonal wavelet, which performs better in frequency band division and signal reconstruction, so it is selected as the mother wavelet $\phi(t)$. The original waveform of VFTO is decomposed into multi-layer wavelets with different frequency layers through the Daubechies wavelet. In this way, the wavelet coefficient matrix $D_i(i = 1, 2, \cdots, k)$ of different frequency layers is obtained to characterize VFTO, where $k$ is the number of maximum frequency layer.

*3.2. SVD of VFTO*

The wavelet coefficient matrixes $D_i$ of different frequency layers of VFTO are arranged in a certain way. In order to improve the effect of extracting VFTO features through SVD, the VFTO feature matrix A is constructed in the form of a Hankel matrix as in Equation (2).

$$A = \begin{pmatrix} D_1 & D_2 & \cdots & D_n \\ D_2 & D_3 & \cdots & D_{n+1} \\ \vdots & \vdots & \ddots & \vdots \\ D_m & D_{m+1} & \cdots & D_k \end{pmatrix} \tag{2}$$

First, n eigenvalues $\lambda_i(i = 1, 2, \cdots, n)$ of the n-order square matrix $A^T A$ are calculated from its characteristic equation, as shown in Equation (3).

$$\left| A^T A - \lambda \right| = 0 \tag{3}$$

Next, n eigenvectors $x_i(i = 1, 2, \cdots, n)$ of the n-order square matrix $A^T A$ are obtained, as shown in Equation (4).

$$\left(A^T A - \lambda_i\right) x_i = 0 \tag{4}$$

These eigenvectors form an n-order matrix, as shown in Equation (5).

$$V = [x_1, x_2, \cdots, x_n] \tag{5}$$

Then, the m-order matrix U is shown in Equation (6).

$$U = [y_1, y_2, \cdots, y_m] \tag{6}$$

$y_i (i = 1, 2, \cdots, m)$ are the eigenvectors of the m-order square matrix $AA^T$. The results of SVD on matrix A are shown in Equation (7).

$$A = U\Sigma V^T \tag{7}$$

In Equation (7), matrixes A, U, V are known. An $m \times n$-order diagonal matrix $\Sigma$ is shown in Equation (8).

$$\sum = diag[\sigma_1, \sigma_2, \sigma_3, \cdots, \sigma_n] \tag{8}$$

In Equation (8), $\sigma_i$ is the singular value of matrix A; it is numerically equal to the square root of the eigenvalue $\lambda_i$ of the n-square matrix $A^T A$.

### 3.3. Singular Value and Energy Characteristic Parameter of VFTO

The VFTO singular value characteristic parameters are constructed based on the each order of singular value $\sigma_i$ of the characteristic matrix. They include the mean value of the singular value $\sigma_{ave}$, the root mean square value of the singular value $\sqrt{\overline{\sigma_i^2}}$, the singular value dispersion coefficient $k(\sigma)$, and the pulse factor of the maximum singular value $I(\sigma_1)$, where $D(\sigma)$ is a singular value variance.

$$\sigma_{ave} = \frac{1}{n} \sum_{i=1}^{n} \sigma_i \tag{9}$$

$$\sqrt{\overline{\sigma_i^2}} = \sqrt{\frac{1}{n} \sum_{i=1}^{n} \sigma_i^2} \tag{10}$$

$$D(\sigma) = \frac{1}{n} \sum_{i=1}^{n} (\sigma_i - \sigma_{ave})^2 \tag{11}$$

$$k(\sigma) = \frac{D(\sigma)}{\sigma_{ave}^2} \tag{12}$$

$$I(\sigma_1) = \frac{\text{Max}[\sigma_1, \sigma_2, \cdots, \sigma_n]}{\sigma_{ave}} = \frac{\sigma_1}{\sigma_{ave}} \tag{13}$$

The energy distribution of each frequency layer obtained by the wavelet decomposition of different types of overvoltage signals is different, so the distribution of the energy value $E_i$ in each frequency layer can be used as an indicator for signal recognition. $E_i$ is shown in Equation (14), where $\alpha_i(j)$ is the $j$-th coefficient of the $i$-th frequency layer wavelet.

$$E_i = \sum_{j=1}^{\infty} |\alpha_i(j)|^2 \tag{14}$$

The energy characteristic parameter $\overline{E_i}$ is the average value of $E_i$ in each band, as follows.

$$\overline{E_i} = \frac{1}{n} \sum_{i=1}^{n} E_i \tag{15}$$

## 4. Simulation Analysis

### 4.1. VFTO Simulation Model

The 550 kV VFTO simulation model was built in ATP-EMTP software, and the time step of the ATP simulation was 1 ns. The model block diagram is shown in Figure 3.

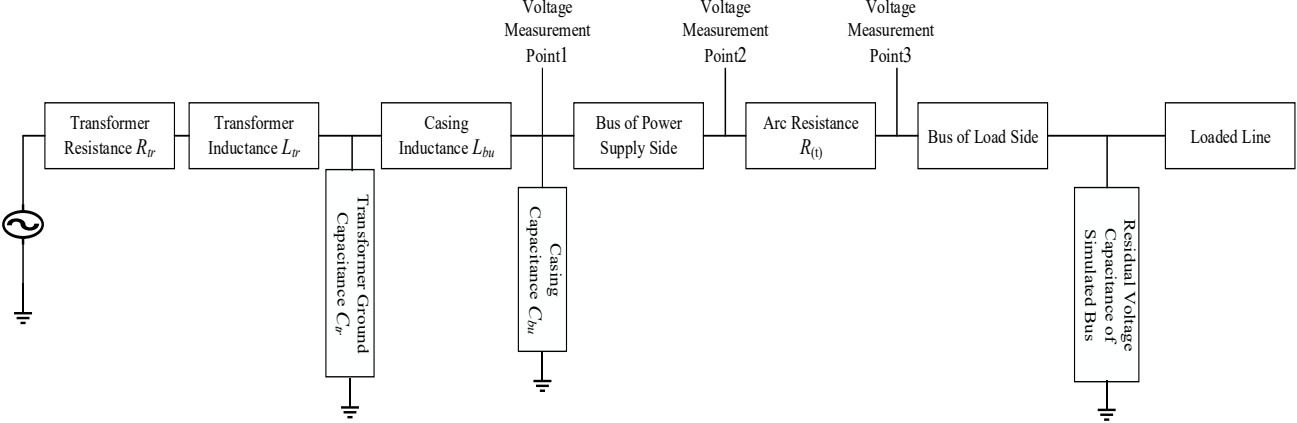

**Figure 3.** VFTO simulation model block diagram.

The arc resistance $R(t)$ is based on a sectional resistance model, as shown in Equation (16).

$$R(t) = R_0 e^{\frac{-t}{T_1}} + r e^{\frac{t}{T_2}} \tag{16}$$

In Equation (16), $R_0 = 10^{12}$ Ω, which is the contact resistance before arcing, and $r = 0.5$ Ω, which is the arc resistance during arcing. $T_1 = 1\,$ns is the time constant of the pre-breakdown process, and $T_2 = 1\,$μs is the time constant of the arc extinguishing process. The curve of $R(t)$ changing with time is shown in Figure 4.

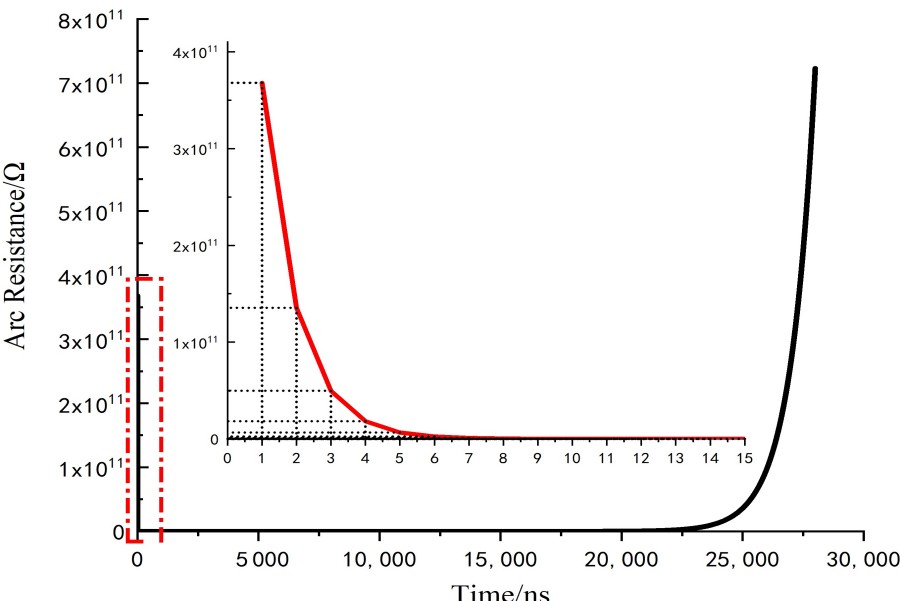

**Figure 4.** Curve of the arc resistance $R(t)$ with time change.

In Figure 4, 0–15 ns is the pre-breakdown processes, and 15–20,000 ns is the stable arcing process, 20,000–30,000 ns and a subsequent session is the arc quenching process. Obviously, the pre-breakdown process is much shorter than the arc quenching process. Bus bars on both sides of the disconnector all adopt the simulation of the distribution parameters in Figure 1; the equivalent inductance $L$ and the equivalent capacitance $C$ of the unit length are shown in Equations (17) and (18).

$$L = \frac{\mu}{2\pi} \ln\left(\frac{R_2}{R_1}\right) \tag{17}$$

$$C = \frac{2\pi\varepsilon}{\ln\left(\frac{R_2}{R_1}\right)} \tag{18}$$

where $\mu = \mu_0 = 4\pi \times 10^{-7}\,\text{H/m}$, $\varepsilon = \varepsilon_r \varepsilon_0 = 8.85 \times 10^{-12}\,\text{F/m}$. The inner radius of the transmission line shell was set to $R_2 = 0.168\,\text{m}$, and the radius of the coaxial high-voltage conductor was $R_1 = 0.07\,\text{m}$ [20]. Thus, the inductance $L = 1.751 \times 10^{-7}\,\text{H}$ and the capacitance $C = 6.352 \times 10^{-11}\,\text{F}$ can be obtained from Equations (17) and (18). Therefore, the wave impedance $Z$ and wave velocity $v$ of the transmission line are shown in Equations (19) and (20).

$$Z = \sqrt{\frac{L}{C}} \approx 52.5\Omega \tag{19}$$

$$v = \pm\frac{1}{\sqrt{LC}} \approx 2.998 \times 10^8\,\text{m/s} \tag{20}$$

Other component parameters are shown in Table 1 [21–24].

**Table 1.** VFTO simulation model component parameters.

| Element | Character | Value |
|---|---|---|
| Transformer resistance | $R_{tr}$ | $1\,\Omega$ |
| Transformer inductance | $L_{tr}$ | 196 mH |
| Transformer ground capacitance | $C_{tr}$ | 5000 pF |
| Casing inductance | $L_{bu}$ | 0.03 mH |
| Casing capacitance | $C_{bu}$ | 320 pF |
| Bus residual voltage | $U(0)$ | $-550$ kV |

### 4.2. Simulation Results

The original time domain waveform of VFTO is shown in Figures 5 and 6.

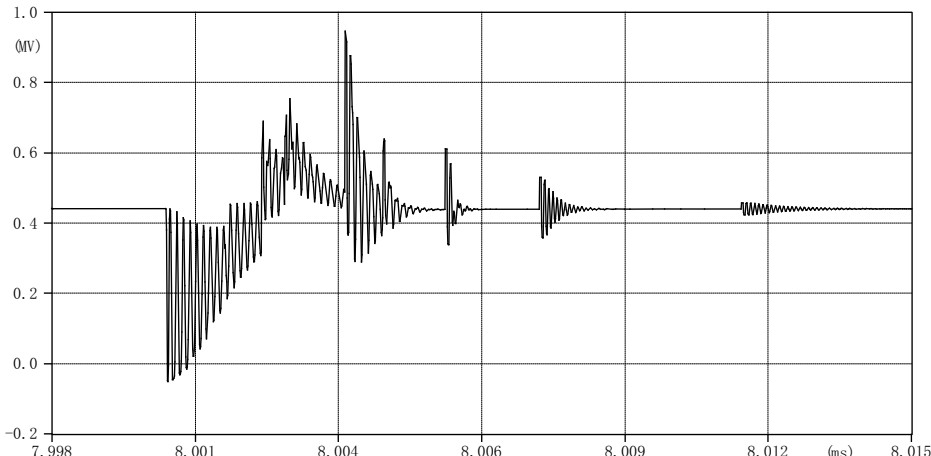

**Figure 5.** VFTO waveform of power supply side.

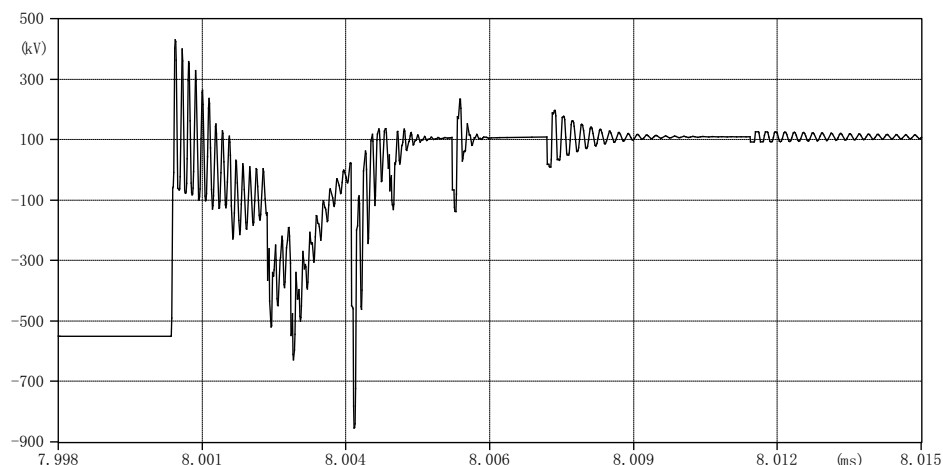

**Figure 6.** VFTO waveform of load side.

The Daubechies wavelet was used to decompose the VFTO waveform of the power supply side into 12 layers. Setting the order of the Daubechies wavelet at $N = 2$, the VFTO waveform of the power supply side was decomposed into the superposition of the approximate coefficient A12 and the 12-layer detail coefficient $D_i$, as shown in Figure 7.

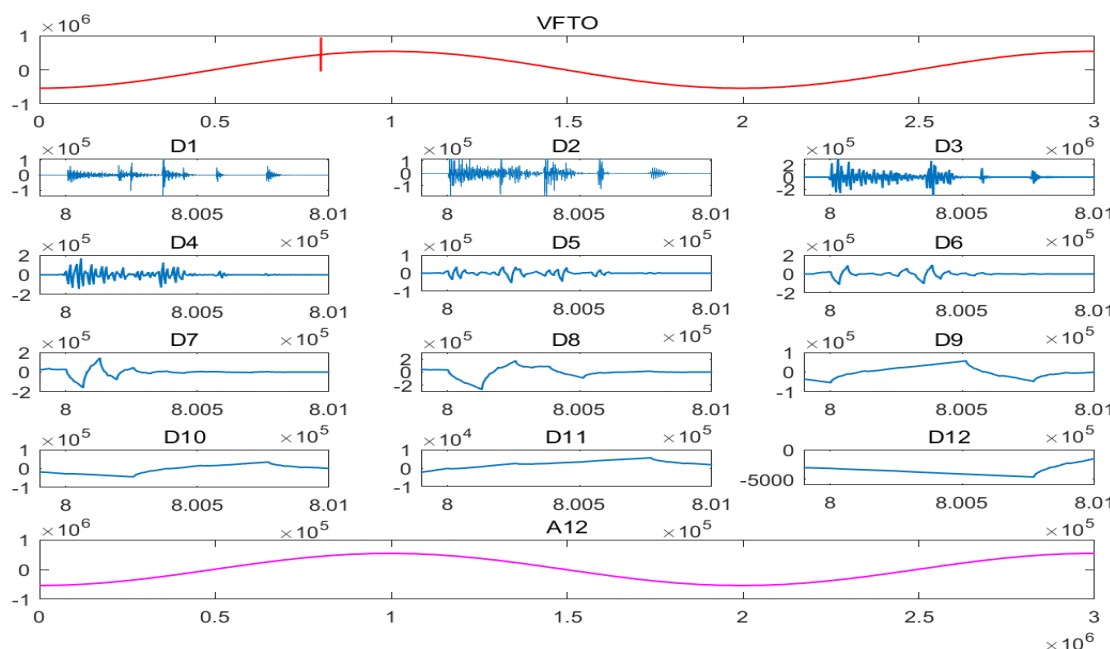

**Figure 7.** VFTO wavelet decomposition results of power supply side.

The detailed coefficients $D_i(i = 1, 2, \cdots, 12)$ of the 12-layer wavelet were obtained by VFTO wavelet decomposition, and the VFTO feature matrix A was constructed according to Equation (2). Then, the SVD of the feature matrix A was carried out to obtain the six-order singular values of VFTO from the power side and the load side, as shown in Figure 8.

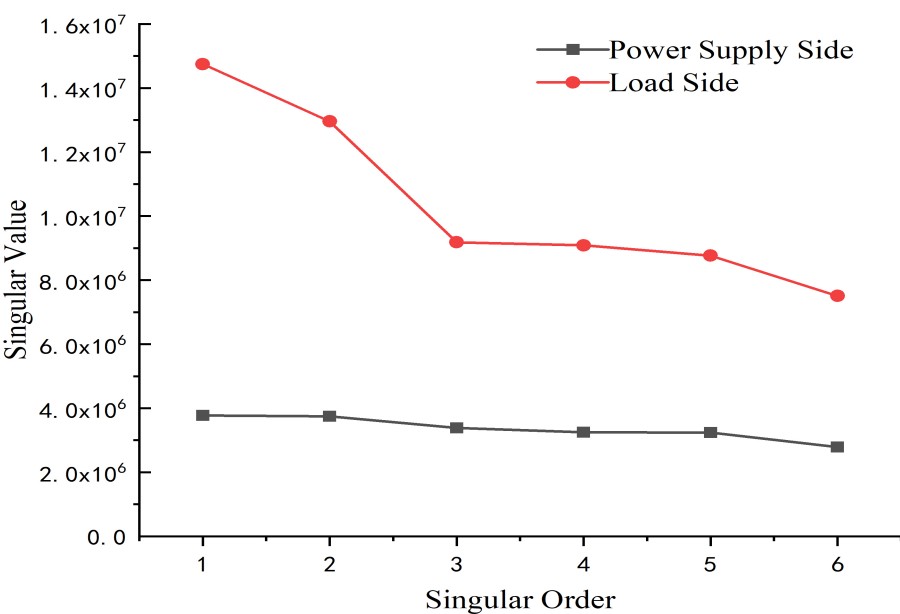

**Figure 8.** Results of SVD of characteristic matrix for VFTO at power side and load side.

The wavelet energy values of each order $E_i$ of the VFTO from the power side and the load side were obtained by simulation, as shown in Figure 9.

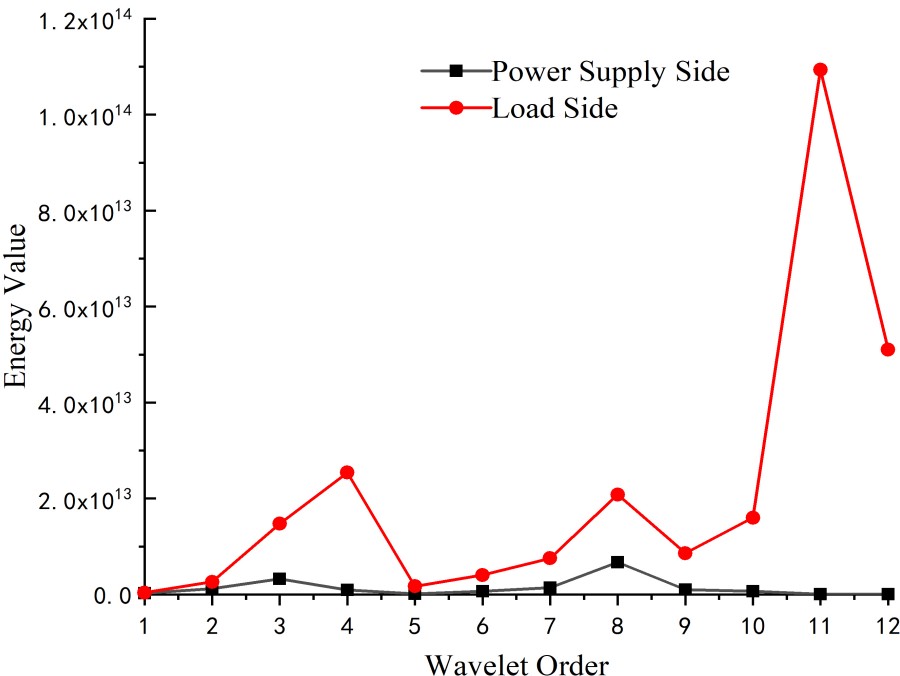

**Figure 9.** Wavelet energy values of VFTO at power side and load side.

*4.3. Comparison with Power Frequency AC Voltage and Other Overvoltages*

Wavelet decomposition and SVD were used on a 550 kV power frequency AC voltage waveform to extract the wavelet coefficients of each frequency layer and calculate the wavelet energy values of each order. The results are shown in Figure 10.

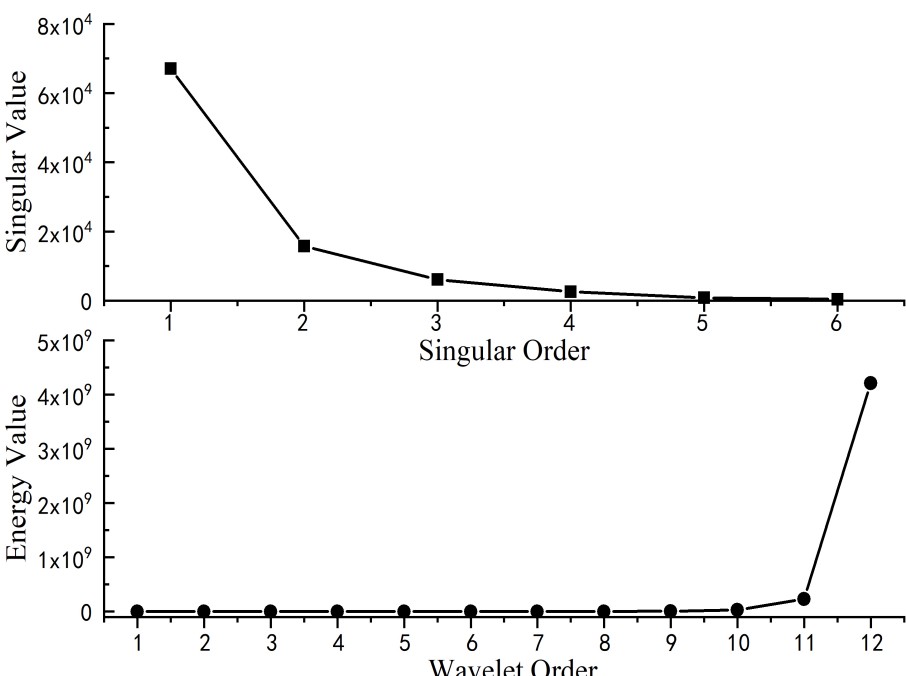

**Figure 10.** Singular values and wavelet energy values of power frequency AC voltage of each order.

Comparing Figures 8–10, it can be found that with the increase of the singular order, the singular values of VFTO on the power supply side and on the load side decrease slowly, while the singular values of power frequency AC voltage decrease rapidly. There are at least two maximum points in the wavelet energy curve of VFTO from the power supply side and the load side, while the power frequency AC voltage wavelet energy curve increases monotonously, and the energy value of the 12th frequency layer is significantly higher than that of other frequency layers. The simulation results of the characteristic parameters of the power frequency AC voltage and the VFTO of the power supply side and load side are shown in Table 2.

**Table 2.** Characteristic parameters of power frequency AC voltage and VFTO.

| Voltage Type | $\sigma_{ave}/10^6$ | $k(\sigma)$ | $\sqrt{\overline{\sigma_i^2}}/10^6$ | $I(\sigma_1)$ | $\overline{E_i}/10^{12}$ |
|---|---|---|---|---|---|
| Power frequency AC voltage | $1.5 \times 10^{-2}$ | 2.4 | $2.83 \times 10^{-2}$ | 4.35 | $3.7 \times 10^{-4}$ |
| VFTO of Power Supply Side | 3.4 | 0.1 | 3.38 | 1.12 | 1.35 |
| VFTO of Load Side | 10.4 | 0.6 | 10.69 | 1.42 | 21.85 |

It can be seen from Table 2 that, compared with the power frequency AC voltage, the mean value of the singular value $\sigma_{ave}$ and root mean square value of the singular value $\sqrt{\overline{\sigma_i^2}}$ of the VFTO increase significantly, and the mean value of the wavelet energy $\overline{E_i}$ of each frequency layer also increases significantly. This indicates that the VFTO has high energy. The pulse factor of the maximum singular value $I(\sigma_1)$ decreases, indicating that the peak characteristic of the first-order singular value weakens. The singular value dispersion coefficient $k(\sigma)$ decreases significantly, indicating that the singular value of each order of the VFTO tends to be concentrated.

The characteristic parameters of the intermittent arc grounding overvoltage (15 kV), lightning overvoltage (900 kV), and ferromagnetic resonance overvoltage (1100 kV) are calculated through the method above. The wavelet energy values of each order of the three types of overvoltage are shown in Figure 11 and these characteristic parameters are shown in Table 3.

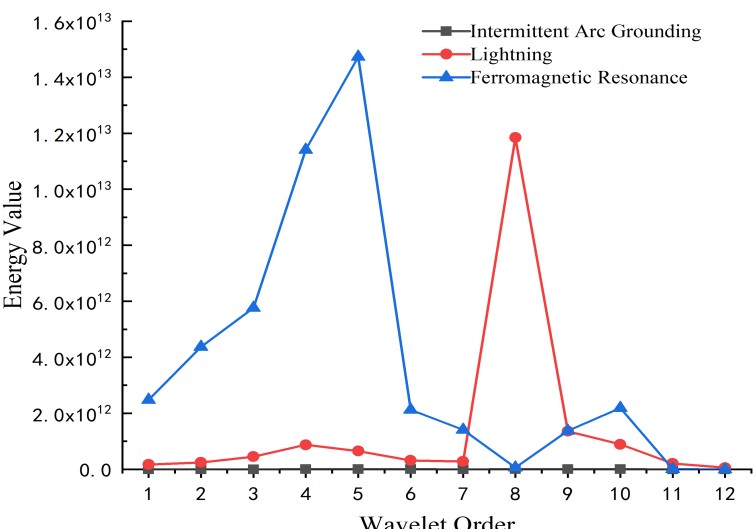

**Figure 11.** Wavelet energy values of intermittent arc grounding, lightning, and ferromagnetic resonance overvoltage.

**Table 3.** Characteristic parameters of three kinds of overvoltage.

| Voltage Type | $\sigma_{ave}/10^6$ | $k(\sigma)$ | $\sqrt{\overline{\sigma_i^2}}/10^6$ | $I(\sigma_1)$ | $\overline{E_i}/10^{12}$ |
|---|---|---|---|---|---|
| Intermittent arc grounding | 0.14 | 0.02 | 0.15 | 1.25 | $2.49 \times 10^{-3}$ |
| Lightning | 3.56 | 0.05 | 3.66 | 1.16 | 1.45 |
| Ferromagnetic resonance | 5.34 | 0.06 | 5.50 | 1.22 | 3.83 |

It can be seen from Tables 2 and 3 that, compared with the power frequency AC voltage, the singular value dispersion coefficient $k(\sigma)$ and pulse factor of the maximum singular value $I(\sigma_1)$ of the three types of overvoltage signal are significantly reduced, indicating that the singular value of each order of overvoltage signal is similar, and the dispersion degree and peak characteristics are suppressed. The voltage level of the lightning overvoltage is about 1.6 times that of VFTO, and the characteristic parameters of the lightning overvoltage are similar to those of VFTO at the power supply side. The ferromagnetic resonance overvoltage is about twice that of the VFTO, and the mean value of the singular value $\sigma_{ave}$ and root mean square value of the singular value $\sqrt{\overline{\sigma_i^2}}$ are about half of those of the VFTO at the load side, and the mean value of the wavelet energy $\overline{E_i}$ of each frequency layer is only 17.5% of that of the VFTO at the load side. Due to the low voltage level, the magnitude of each characteristic parameter value is low. Due to low voltage level, the order of magnitude of each characteristic parameter of the intermittent arc grounding overvoltage is low.

In a real system, the original waveform of overvoltage on the side of the isolation switch is measured by the oscilloscope first, and then the wavelet transform and singular value decomposition are carried out by a computer to decompose the original waveform. Finally, the value of each characteristic parameter is calculated and compared with that of the VFTO. If the error is less than the threshold, the overvoltage measured can be considered as VFTO.

## 5. Conclusions

In order to accurately identify VFTO, we obtained the original waveform of VFTO through ATP-EMTP software simulation. After the process of wavelet decomposition, wavelet coefficient extraction, construction of the wavelet coefficient matrix, and SVD, four singular value recognition characteristic parameters and one energy recognition characteristic parameter were constructed. The value of each characteristic parameter of VFTO was compared with that of power frequency AC voltage and three other types of

overvoltage, verifying the effectiveness of the proposed VFTO identification method based on WT and SVD. This method can be theoretically applied to different transient processes in other different types of substations, such as symmetrical and unsymmetrical faults and circuit breaker openings.

**Author Contributions:** Conceptualization, G.X., Q.R., M.Y., P.X., Q.C., J.F., H.G. and H.W.; methodology, G.X., Q.R., M.Y., P.X., Q.C. and J.F.; software, G.X., Q.R., M.Y., P.X., Q.C. and J.F.; validation, H.G. and H.W.; formal analysis, G.X., Q.R., M.Y., P.X., Q.C. and J.F.; investigation, J.F., H.G. and H.W.; resources, G.X.; data curation, G.X., Q.R., M.Y. and P.X.; writing—original draft preparation, G.X., Q.R., M.Y. and P.X.; writing—review and editing, P.X., M.Y. and H.W.; visualization, G.X., Q.R., M.Y. and P.X.; supervision, Q.C., J.F., H.G. and H.W.; project administration, G.X.; funding acquisition, P.X. All authors have read and agreed to the published version of the manuscript.

**Funding:** This research was funded by Central Universities (WUT:2020IVA031).

**Institutional Review Board Statement:** Not applicable.

**Informed Consent Statement:** Not applicable.

**Data Availability Statement:** Not applicable.

**Conflicts of Interest:** The authors declare no conflict of interest.

## Abbreviations

The following abbreviations are used in this manuscript:

VFTO   Very Fast Transient Overvoltage
GIS    Gas Insulated Substation
WT     Wavelet Transform
SVD    Singular Value Decomposition

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
