# Peer review of "Research on VFTO Identification of GIS Based on Wavelet Transform and Singular Value Decomposition"

_energies, doi:10.3390/en15093367_

Round 1
Reviewer 1 Report
The paper presents an interesting application of Digital Signal processing for detecting very fast transient overvoltages. The content is interesting and represents a good approach validated through simulations. However, the paper requires a major editing in style and grammar. Also, several other topics are omitted or unnecessarily mentioned in the paper, which as described as follows:
A nomenclature table needs to be added to the paper, it contains many acronyms that are not even introduced in the paper. For example, Gas Insulated Switchgear can easily be understood as Geographic Information system, which is a very common acronym used these days in power engineering.
The introduction focuses on the experiment but not on why this is a topic of interest. The authors dedicate only 6 lines to describe the problem with no background. Then, they dedicate the rest of the introduction to talk about their findings. I consider that a proper introduction to the problem needs to be presented before talking about the experiments conducted for dealing with it.
The paper in general demands a revision of style and grammar. The way sentences are put together makes it look robotic. Please use connectors and transitions for smoothing the ideas presented and to demonstrate that they are connected instead of a list of separate topics. Also, review the time tense since several sentences are presented in present tense when talking about past events.
Why did the authors use the Daubechies wavelet pattern? Is there any difference if they decided to use a “Mexican hat” for example? What properties of the selected pattern make it suitable for this experiment?
The authors state in the abstract that “The VFTO Identification method based on WT and SVD is practical”, however, they did not provide proof of it. The results presented in the paper are based on simulations, no proof of physical implementation or Real-time testing was presented in the paper to support the affirmation mentioned above. If there is no proof of the physical implementation of the suggested analysis method, the statement above should be removed from the paper.
Reviewer 2 Report
This paper studies the fast transient overvoltage identification of GIS based on wavelet transform and singular value decomposition. Some comments should be addressed.
-An updated literature review should be conducted. The relevance to Energies should be enhanced with the considerations of scope and readership of the Journal.
-The contributions of this paper are suggested to be reorganized to improve the clarity and highlight the novelty.
-The comparison with the state-of-the-art methods in literature should be provided. In addition, it is suggested to validate the scalability of the proposed method in a large test network.
-The authors should explain how can be implemented in the real system.
Reviewer 3 Report
The authors have provided a research on VFTO identification by means of simulation with ATP and Matlab. The paper is well-written and deserves to be published, but the authors should address the following comments:
1) which is the time step of the ATP simulation?
2) Which is the computational effort of the total procedure?
3) Please provide some references for the values adopted in Section 4
4) Improve the quality of Figures 3 and 6
5) When you deal with lightning, you are implicitly speaking about direct events. How could your method be applied to indirect events?
6)I suggest to add a flowchart explaining the simple operation principle of GIS in Section 2.
Reviewer 4 Report
In case of a Smart Grid, addressing the transient events are highly critical since the definition of smart grid itself suggests that it should be self healing.
In those cases, devising new methodologies for very fast transient overvoltage is highly important. In this case the authors have applied the wavelet transform and singular value decomposition during the disconnection of the Gas Insulated Substation from the main system.
Good job done by authors. I have very few quick questions.
1) Are there any other techniques apart from wavelet transform and singular value decomposition?
2) Is there any particular reason to choose these methods?
3) Can it be applied to all types of substations or only Gas insulated substations when disconnected during such an event?
4) Can the methodology be applied to other transient events like symmetrical and unsymmetrical faults, circuit breaker opening?
5) Have you ever tried this with different systems and have you observed the behavior pattern with different systems.
I am recommending the paper towards acceptance after major revisions.
Round 2
Reviewer 1 Report
Even though several improvements have been applied to the paper structure and style, there are still several mistakes that need to be corrected. For example, in line 31, the authors start the paragraph with “Many scholars have…”, the question here is, how much is “many”. Please, avoid vague statements in a scientific paper. Going back to the same line, instead of writing “Many scholars have conducted in-depth …” I suggest writing “There is growing interest in…”. The same type of mistakes can be found across the paper.
I find the writing style very robotic, sometimes, it feels like there is no connection at all between sentences located one after the other. In the conclusions (and across the paper), please remove “etc.” again, avoid vague terms/statements.
Reviewer 2 Report
All the comments have been addressed.
Reviewer 4 Report
The authors have satisfied the queries asked. Good job.The paper can now be accepted.
Round 3
Reviewer 1 Report
Good job.